# A Review of *Polygonatum* Mill. Genus: Its Taxonomy, Chemical Constituents, and Pharmacological Effect Due to Processing Changes

**DOI:** 10.3390/molecules27154821

**Published:** 2022-07-28

**Authors:** Lu Luo, Yixing Qiu, Limin Gong, Wei Wang, Ruiding Wen

**Affiliations:** 1TCM and Ethnomedicine Innovation and Development International Laboratory, Innovative Materia Medic Research Institute, School of Pharmacy, Hunan University of Chinese Medicine, Changsha 410208, China; lulu1209jinx@163.com (L.L.); qiuyixing@hotmail.com (Y.Q.); wruiding@163.com (R.W.); 2School of Chinese Medicine, Macau University of Science and Technology, Macau 999078, China

**Keywords:** genus *Polygonatum*, classification, chemical composition, processing, pharmacological effect

## Abstract

Ethnopharmacological relevance: The genus *Polygonatum* Tourn, ex Mill. contains numerous chemical components, such as steroidal saponins, polysaccharides, flavonoids, alkaloids, and others, it possesses diverse pharmacological activities, such as anti-aging, anti-tumor, immunological regulation, as well as blood glucose management and fat reducing properties. Aim of the review: This study reviews the current state of research on the systematic categorization, chemical composition, pharmacological effects, and processing changes of the plants belonging to the genus *Polygonatum*, to provide a theoretical foundation for their scientific development and rational application. Materials and methods: The information was obtained by searching the scientific literature published between 1977 and 2022 on online databases (including PubMed, CNKI, SciFinder, and Web of Science) and other sources (such as the *Chinese Pharmacopoeia* 2020 edition, and Chinese herbal books). Results: The genus *Polygonatum* contains 79 species, and 233 bioactive chemical compounds were identified in them. The abundance of pharmacological activities, such as antioxidant activities, anti-fatigue activities, anti-inflammatory activities, etc., were revealed for the representatives of this genus. In addition, there are numerous processing methods, and many chemical constituents and pharmacological activities change after the unappropriated processing. Conclusions: This review summarizes the taxonomy classification, chemical composition, pharmacological effects, and processing of the plants belonging to the genus *Polygonatum*, providing references and research tendencies for plant-based drug development and further clinical applications.

## 1. Introduction

The genus *Polygonatum* belongs to a perennial herbaceous plant whose English name is King Solomon’s seal, and it belongs to the Asparagaceae family. There are about 79 species of *Polygonatum* in the globe, which are extensively distributed in the northern hemisphere. About 39 species are recorded growing in China [1]. The genus *Polygonatum* has long been valued for its medicinal, diet, and healthcare values, the rhizomes are medicinal portions [2]. *Polygonatumi rhizoma* and *Polygonatumi odorati rhizoma* belong to the genus *Polygonatum* and have been added to the “*Chinese Pharmacopeia*” (2020 edition) [3]. The genus *Polygonatum* contains polysaccharides, flavonoids, steroids, coumarins, and other chemical components [4]. As a medicinal plant, its dried rhizome has anti-aging, anti-oxidation, immune regulation, anti-inflammatory, and anti-cancer effects, and is clinically used to treat fatigue, weakness, diabetes, cough, and loss of appetite [5]. However, the unprocessed herbs in genus *Polygonatum* can irritate the throat, the raw rhizomes of *Polygonatum* Mill. are processed by repeated steaming and drying (nine times each) in order to reduce toxic components, and improve their primary functioning, taste, and pharmaceutical effects [6].

Previously, reviews that focused on some species, *P. odoratum*, *P. cyrtonema*, *P. kingianum*, and *P. sibiricum* have been conducted. To the authors’ knowledge, no study has reviewed the taxonomy classification, chemical composition, pharmacological effect, and processing of the whole genus. This review is aimed to critically evaluate available research reports on the genus, and systematically organize and present the findings.

## 2. Classification of *Polygonatum* Mill.

The genus *Polygonatum* comprises 79 species. Among them, 39 species distributed in China were recorded in the Chinese monograph “*Flora of China*” [1], and the other 40 species were included in the World Checklist of Selected Plant Families (WCSPF, World Checklist of Selected Plant Families: Royal Botanic Gardens, Kew). The contributions and the first recorded time of the species are summarized (Table 1).

## 3. Chemical Constituents of *Polygonatum*

As mentioned in the introduction, the herbs in the genus *Polygonatum* contain many chemical components, such as steroidal saponins, polysaccharides, flavonoids, and alkaloids. The author summarized 233 compounds isolated from this genus from 1977 to 2022, which contained 124 steroidal saponins, 68 flavonoids, triterpenoid saponins, 16 alkaloids, 3 quinones, and 6 lignans.

### 3.1. Steroidal Saponins

Steroid saponins are formed by the condensation of steroid sapogenins and sugar. The carbon frame of steroid sapogenins is made up of 27 carbon atoms and is based on spirostane. According to the configuration of C_25_ in the spirostane structure and the cyclization state of the F ring, it is divided into spirostanol, isospirostanol, furostanol, and pseudo spirostanol types. The main pharmacological active substances are the first three types of steroidal saponins in the genus *Polygonatum*. The glycosyl moiety (mainly glucose, galactose, xylose, rhamnose, and fucose) is an important factor in the formation of the molecular diversity of the genus *Polygonatum* saponins. The details of the compounds are shown in Table 2, and the structural formulas are shown in Figure 1, Figure 2, Figure 3 and Figure 4.

### 3.2. Flavonoids

Flavonoids originally refer to the general term for compounds derived from 2-phenylchromone. It generally refers to a set of compounds formed by two benzene rings connected through three carbon atoms, a general term for a series of compounds with a C_6_-C_3_-C_6_ structure. Flavonoids mostly include flavones, flavonols, dihydroflavonoids, isoflavones, and homoisoflavonoids in *Polygonatum* Mill. (Table 3, Figure 5).

### 3.3. Triterpenoid Saponins

Triterpene saponin is a class of glycosides in which aglycones are triterpenoid compounds, mainly distributed in terrestrial higher plants. Triterpenoids are a type of terpenoids. Their basic core skeleton is made up of 30 carbon atoms. They exist in plants in three forms; free, in the form of glycosides, or esters combined with sugars. The main active ingredients of many well-known Chinese herbal medicines, such as *Ginseng*, *Glycyrrhiza uralensis*, and *Anemarrhena asphodeloides*, have triterpene saponins. Some saponins also have valuable biological activities, such as antibacterial activity, sedation, and anticancer. The triterpenoid saponins isolated and identified from the plants of the genus *Polygonatum* are shown in Table 4, and structures are shown in Figure 6.

### 3.4. Alkaloids

Alkaloids are nitrogen-containing alkaline organic compounds in nature (mainly in plants, but some also exist in animals). They have a complex ring structure, and nitrogen is usually contained in the ring. It has significant biological activity and is one of the most effective ingredients in Chinese herbal medicine. *Polygonatum* has a low content of alkaloids and a changeable structure. Alkaloids have been found in *P. odoratum*, *P. kingianum*, *P. cirrhifolium*, *P. verticillatum*, and *P. alte-lobatum* (Table 5, Figure 7).

### 3.5. Quinones

There are now three quinones isolated from *P. odoratum* and *P. alteolobatum* [13,47]. Ubiquinones with a benzoquinone structure can engage in the redox process in vivo and are a family of coenzymes considered to coenzyme Q in biological oxidation reactions. It has significant therapeutic medical value and can be used to treat cardiovascular disease, hypertension, and cancer. The tree quinones are emodin-8-O-β-D-glucopyranoside (a), polygonaquinone A (b), and polygonaquinone B (c), and their structures are in Figure 8.

### 3.6. Lignans

Lignans exist in plants and belong to a kind of phytoestrogen that has antioxidation functions. Ru [56] isolated four lignans from *P. sibiricum* for the first time, which were (+)-syringaresinol, (+)-syringaresinol-O-β-D-glucopyranoside, liriodendrin, (+)-pinoresinol-O-β-D-glucopyranosyl-(6→1)-β-D-glucopyranoside. Gao [39] also found liriodendrin from the fresh *P. sibiricum* rhizome. Chen Hui et al. [60] published three lignans from the ethyl acetate layer of *P. sibiricum* rhizomes, namely (+)-syringaresinol, 5-hydroxy-7-methoxy-4,6-dimethyl- 2-benzofuranone, terpineol.

### 3.7. Polysaccharides

Polysaccharide is one of the main active ingredients of the genus *Polygonatum*. Due to the complexity of the structure and the relatively large molecular weight of *Polygonatum* Polysaccharide, there are relatively few studies on the chemical structure. At present, galactomannan galactose has two types of neutral polysaccharides (PSB-2A, PSB-1B), two types of acid polysaccharides (PSW-2A-1, PSW-3A-1), two glycoproteins (PSW-4A, PSW-5B), and neutral galactose (PSW-1B-b), separated and purified from the rhizome extract of *P. sibiricum* [61]. Different extraction methods result in different monosaccharide compositions. The structure of the original *P. cyrtonema* polysaccharide was composed of arabinose, galactose, glucose, and xylose with a molecular ratio of 1.34:7.42:54.47: 36.95 by Wu [62], and other groups also proved that cellulase-assisted extraction and hot water extracted polysaccharide of polysaccharides from *P. odoratum* consisted of mannose, glucosamine, rhamnose, glucose, galactose, and arabinose, with a molecular ratio of 7.80:1.08:1.63:65.93:3.58:1.00 and 11.22:0.23:0.23:17.59:2.73:9.10, respectively [63]. Interestingly, this article did not explain the specific temperature of hot water. Both 50 °C and 90 °C were hot water. Different species have different monosaccharide compositions. Zhao [64] proved that polysaccharides from *P. sibiricum*, *P. cyrtonema*, and *P. kingianum* were mainly composed of fructose, galacturonic acid, and galactose, with small amounts of rhamnose, arabinose, xylose, and glucose; while polysaccharides from *P. odoratum* mainly consisted of fructose with trace amounts of galacturonic acid, galactose, rhamnose, arabinose, xylose, and glucose.

## 4. Pharmacological Activities

### 4.1. Antioxidant Activities

The *P. sibiricum* (PSP) may modulate the Klotho-FGF23 endocrine axis, reduce oxidative stress, and maintain calcium and phosphorus metabolism balance [65]. *Polygonatum cyrtonema* polysaccharide (PCP) significantly increased superoxide dismutase (SOD) and glutathione peroxidase (GSH-Px) activities and decreased malondialdehyde (MDA), indicating PCP could increase antioxidant enzyme activity to protect against lipid peroxidation and oxidative stress induced by exhaustive exercise. Additionally, PCP dramatically increased the protein levels of bone morphogenetic protein-2 (BMP-2), phosphor-Smad1, Runt-related transcription factor 2 (Runx2), and osteocalcin (OC). These findings revealed a link between PCP’s antioxidant property and its anti-fatigue function [66]. By decreasing oxidative stress, oral treatment of PSP may mitigate the aging and damage generated by D-galactose in the heart. D-gal treatment decreased reactive oxygen species (ROS) and MDA and enhanced SOD levels in the hearts of mice. By reducing the levels of 8-hydroxydeoxyguanosine (8-OHdG) and 4-hydroxy-2-nonenal, PSP also prevented oxidative stress-induced DNA damage and lipid peroxidation (4-HNE) [67]. Regarding other species, extracts of *P. alte-lobatum* (EPA) dose-dependently reduced exercise-induced urea nitrogen and malondialdehyde and enhanced hepatic glycogen, an essential workout fuel [68]. In addition, the surface structure of PSP was smooth and irregular, and bead-like structures were identified, suggesting that PSP could be employed for encapsulating purposes in the design of drug delivery systems [69]. In other research, PSP was used as a stabilizer to fabricate SeNPs (selenium nanoparticles) under a simple redox system. The ability of SeNPs to get rid of free radicals was greatly improved by adding PSP to the surface of the nanoparticles [70].

### 4.2. Anti-Fatigue Activities

The trend analysis showed that EPA supplementation improved endurance running time 1.62-fold. EPA boosted rats’ endurance time to exhaustion, showing it may increase exercise tolerance [71]. Swimming time was used to test the anti-fatigue activity of PCP. Dose- and age-dependent increases in fatigue time were seen after PCP treatment, indicating that PCP may enhance the endurance of mice during exercise. A significant correlation was found between exhaustive swimming duration and osteocalcin levels in mouse muscle fibers treated with PCP, showing that PCP’s anti-fatigue effect is linked to energy metabolism and osteocalcin signaling [72].

### 4.3. Anti-Inflammatory Activities

The anti-inflammatory mechanism of *P. sibiricum* that suppressed the production of pro-inflammatory mediators and was linked to the downregulation of the NF-B pathway was discovered [73]. In vitro anti-inflammatory effects of *P. verticillatum* were positively correlated with the total phenolic content, flavonoid content, and condensed tannin content. It showed that *P. verticillatum* had powerful antioxidant, anti-inflammatory, and cancer-preventing properties caused by the plant’s secondary metabolites [74]. Using reverse transcription-quantitative PCR and western blotting, PSP decreased body weight, blood lipids, blood glucose, insulin, resistin, adiponectin, and abdominal fat pad weight. It also reversed abnormal expression levels of inflammatory factors and lipid metabolism genes [75].

### 4.4. Antihypoglycemic Activities

*Polygonatum* Mill. has been used as herbal medicine to treat type 2 diabetes mellitus (T2DM). The polysaccharides of *Polygonatum rhizoma* were analyzed for their structure and bioactivity. At concentrations between 1.0 and 10.0 mg/mL, polysaccharides from *Polygonatum rhizoma* showed varied levels of hypoglycemic action in a dose-dependent manner [76]. The active ingredients are not just polysaccharides but also saponins. The total saponins extract from *P. sibiricum* could inhibit α-amylase and α-glucosidase, a in insulin resistant (IR) -HepG2 cells model [77]. In the same activity as other species, polysaccharides of *P. kingianum* increased the expression of insulin receptor substrate-1 (IRS-1), phosphoinositide 3-kinase (PI3K), and protein kinase B (AKT), showing that polysaccharides of *P. kingianum* adjust glucose metabolism by activating the PI3K/AKT signaling pathway.

### 4.5. Immunological Activities

The vitality of macrophages is a measure of immune activation and activator cytotoxicity [78]. In a dose-dependent way, PSP caused dendritic-like morphological alterations in RAW 264.7 cells and enhanced the production of nitric oxide, TNF-α, and IL-6. The expression of iNOS, COX-2, NF-kB, and phosphorylated p38 MAPK was increased in RAW 264.7 cells treated with PSP [79]. Different concentrations of extractants have different effects. The *P. sibiricum* ethanol 75 (PSE75) increased the mRNA expression of Th1 and Th2 molecular markers compared to *P. sibiricum* ethanol 30 (PSE30). Immunoglobulins G and M were substantially higher in PSE75 than in PSE30. The immunological regulatory action of PSE75 may be mediated by a change in the makeup of gut microbes [80]. In another study, PSP increased the expression of IL-2 and TNF-α in lymphocytes of the spleen. In addition, PSP therapy increased the dose-dependent recovery of natural killer cell activity [81]. The same as other species, *P. odoratum* polysaccharides (POP) also exhibit immunomodulatory activity [82]. Immunomodulation, infection prevention, gut environment enhancement, and cancer suppression of the *Polygonatum* genus have been studied extensively.

### 4.6. Other Activities of Polygonatum Mill.

*P. kingianum* polysaccharides (PKP) and *P. kingianum* aqueous extract (PKAE) alleviated uranium-induced cytotoxicity by regulating mitochondria-mediated apoptosis and the GSK-3β/Fyn/Nrf2 pathway [83]. PCP exerted antidepressant effects by regulating the oxidative stress-calpain-1- NOD-like receptor protein 3 (NLRP3) signaling axis. PCP prevented chronic unpredictable mild stress-induced changes in the calpain system and reduced depression-like behavior [84]. Moreover, methanol extract from *P. odoratum* administration reversed intestinal microbiota compositions, inhibiting H_2_S-related bacteria, a lower level of H_2_S, and higher content of short-chain fatty acid-related bacteria [85]. PSP also can act as a prebiotic, regulating the intestinal tract probiotics. At the phylum level, PSP treatment raised the number of *Lactobacillus* and decreased the abundance of *Lachnospiraceae* and *Bacteroides* (at the genus level). The make-up of microbes shifted. The PSP group increased SCFAs, such as acetic acid, propionic acid, and butyric acid than the control mice [86].

## 5. Processing of *Polygonatum* Mill.

### 5.1. Processing Methods of Polygonatum Mill.

There were many methods of processing the genus *Polygonatum* in the past to increase the curative effect and reduce toxicity. Calcium oxalate monohydrate (COM) raphides may be some of the irritating components of the genus *Polygonatum*. After processing, there were far fewer COM raphides. The raphide bundles that remained adhered together and were difficult to separate and most single raphides were disintegrated, particularly at their tips [87]. Some scholars believe that volatile components, such as n-hexanal and camphene, are also irritating components of the genus *Polygonatum* [88]. There are big differences in the processing and use of traditional Chinese medicine. According to the records of relevant documents in various regions, the processing methods of *Polygonatum* plants include steaming, wine steaming, and wine stewing. There are big differences in the auxiliary materials [89]. The most commonly used methods are steaming, wine steaming, and wine stewing. The *“Chinese Pharmacopeia”* includes wine steaming and stewing [3]. Whether steaming or stewing can achieve the purpose, using wine as an auxiliary material can increase the dissolution of certain compounds [90]. The author summarizes all methods of processing the genus *Polygonatum.* (Table 6).

### 5.2. Effect of Processing on the Chemical Composition

Polysaccharides are one of the main components of the medicinal *Polygonatum* Mill., which changes after processing. During the nine steaming and nine drying processes of the genus *Polygonatum*, with the increase in steaming times, the polysaccharide content first decreases and then stabilizes. Baolai Fan [96] analyzed polysaccharide component changes in distilled and processed *P. cyrtonema* by PMP(1-phenyl-3-methyl-5-pyrazolone) pre-column derivatization, and high-performance liquid chromatography–mass spectrometry/mass spectrometry (HPLC–MS/MS) technology; the processed *Polygonatum* polysaccharide is mainly composed of galactose and mannose, followed by glucose. Moreover, other groups [97] showed that as the number of repetitions of steaming increases, polysaccharides gradually decompose into small monosaccharides. For these monosaccharides, the content after four steaming seems relatively stable [98]. All these dynamic changes in polysaccharides and monosaccharides result from the decomposition of glycosidic bonds that steaming can destroy. Others [99] revealed that the content of 5-hydroxymethyl furfural, galactose, and glucose increased after the fourth steaming and tended to be stable. Moreover, the raw rhizome’s strong numb tongue taste decreased progressively until disappearance after the fourth steaming, and the sweet taste gradually turned from slight to strong at the fourth steaming, which indicated that the toxic components were greatly reduced and the flavor was greatly developed at the fourth steaming.

During the nine steaming and nine drying processes of *P. cyrtonema*, the content of saponins increased first and then stabilized with the increase of steaming and drying. Since diosgenin is a prerequisite for many other saponins, some researchers have found that the content of diosgenin in *P. cyrtonema* after the wine is lower than that of raw products [100].

### 5.3. Influence of Processing on Pharmacological Effects

#### 5.3.1. Antioxidant Activities after Processing

There are various processing methods for evaluating the antioxidant activity of *P. odoratum* flavones and determining which procedure could preserve such activity. The yeast fermentation had the least effect on the antioxidant activity of *P. odoratum* flavones, making it the optimal way of food processing for *P. odoratum*. In contrast, extrusion and high-pressure treatment marginally diminished the flavones’ antioxidant activity [101]. The same is true for *P. odoratum* flavones, and the fermentation method evaluated the antioxidant properties of flavones extracted from fermented *P. odoratum* samples. *Lactobacillus*, yeast, and *Aspergillus* fermentation were examined. By fermenting with *Lactobacillus* and yeast, the antioxidant capacity of *P. odoratum* flavones was found to be diminished. Fermentation with *Aspergillus niger* enhanced the antioxidant capacity of *P. odoratum* flavones [102]. The flavones are not the only compounds that have antioxidant activity. Using radical scavenging experiments, the antioxidant activity of PSP was evaluated. It was discovered that the radical scavenging activity of PSP was significantly enhanced after steaming and increased steadily with increasing numbers of steaming processes [99]. Although the polysaccharides content was decreased after steam-processing, antioxidant and hypoglycemic activities of *P. cyrtonema* were enhanced [103].

#### 5.3.2. Anti-Fatigue Activities after Processing

Polysaccharides in the processed products of *P. cyrtonema* were the active compounds against exercise tiredness, which were more active in the plant’s processed products than in its raw materials. It offers anti-fatigue benefits for swimming exhausted mice, liver glycogen content rose, and the impact of the processed product was superior to that of the raw materials [104,105].

#### 5.3.3. Anti-Inflammatory Activities after Processing

Lung damage caused by LPS may be treated with PSP polysaccharides and its honey-processed polysaccharides, both of which include anti-inflammatory properties. The honey-processed polysaccharides had a greater anti-inflammatory impact than raw materials polysaccharides, which inhibited the synthesis and release of IL-1, IL-6, and TNF-α [106]. In another study, raw polysaccharides and nine-steam-nine-bask processing *P. cyrtonema* demonstrated no toxicity and side effects on lipopolysaccharide-induced RAW264.7 cells and showed obvious inhibitory effects on the inflammatory cytokines NO, TNF-α, IL-1β, IL-6, and MCP1 in a dose-dependent manner. Thus, it is assumable that polysaccharides from raw materials and nine-steam-nine-bask processing *P. cyrtonema* play an anti-inflammatory role by inhibiting the expression of related inflammatory factors [107].

#### 5.3.4. Anti-Hypoglycemic Activities after Processing

The different extractions parts from the crude and steam-processed *P. cyrtonema* were tested for inhibiting α-glucosidase activities from exploring potential active sites [108]. The result shows that the inhibition rate of the ethyl acetate phase of the steamed product reached 87.21%, IC50 = 1.369 mg/mL, and the inhibition rate of the ethyl acetate phase of the raw product reached 59.38%, indicating that the active ingredient in the ethyl acetate phase of the steamed product has a strong effect on α-glucose. Li [109] found that fermented *P. sibiricum* ameliorated the lipid accumulation in liver and white adipose tissue by inhibiting lipogenesis, enhancing lipolysis, and fatty acid oxidation. Therefore, it lowered the fasting blood glucose, insulin, total cholesterol, and triglyceride. In addition, it could reduce glycated hemoglobin in the homeostasis model after P. sibiricum was fermented. When P. sibiricum was processed using the traditional technology of “Nine-Steam-Nine-Bask”, its 70% ethanol extracts exhibited the relief of glycolipid metabolism abnormalities in type 2 diabetic mice [110].

#### 5.3.5. Immunological Activities after Processing

As mentioned above, PCP content was considerably reduced by steaming. Compared to PCP from the raw rhizome, the immunological activities of PCP after 2 and 4 h were greater on PCP. The longer the steaming duration (6–12 h), the more PCP was destroyed, which had a detrimental effect on the immune system [111]. In another study [112], IL-2, IL-6, TNF-α, and IFN-α secretions reversed to normal levels after treatment with the water-soluble PSP extracted from crude and wine-processed PSP in the immunosuppressive model for spleen-deficient mice. PSP that had been wine-processed had more immunological effects than PSP from crude. The steam-processed PSP might be linked to the regulation of the JAK1-STAT1 pathway and the elevation of hematopoietic cytokines (erythropoietin, granulocyte colony-stimulating factor, TNF-a, and IL-6). It could also significantly increase peripheral blood cells, restore the splenic trabecular structure, and bring immune cytokines back to normal levels [113].

## 6. Conclusions

In conclusion, based on the current state of research, *Polygonatum* Mill. belongs to a renewable resource herb with many species. It has numerous chemical components and pharmacological activities. Various research studies have been conducted to evaluate the traditional uses of the genus *Polygonatum*, and all of the research supports the traditional claims. The authors believe that corresponding standards of *Polygonatum* Mill. should be established according to their various clinical applications first [6]. Secondly, an abundance of traditional uses has not been evaluated, especially in species other than *P. sibiricum*, *P. cyrtonema*, *P. kingianum*, and *P. odoratum*. Hence, further research is needed to exploit the many uses of the *Polygonatum* species. The final objective should be to research the usefulness of parts on the ground and fibrous roots to ensure effective protection and the sustainable development of resource applications.

## Figures and Tables

**Figure 1 molecules-27-04821-f001:**
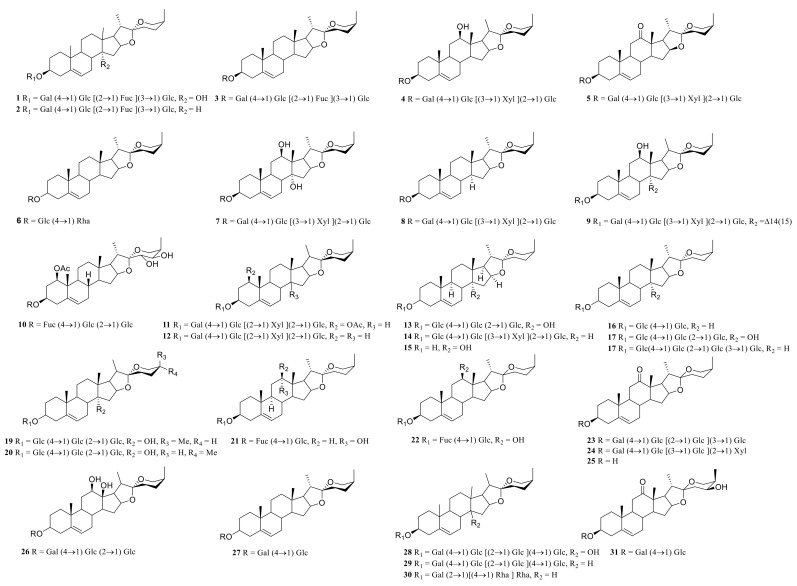
Structures of spirostanol from *Polygonatum* Mill.

**Figure 2 molecules-27-04821-f002:**
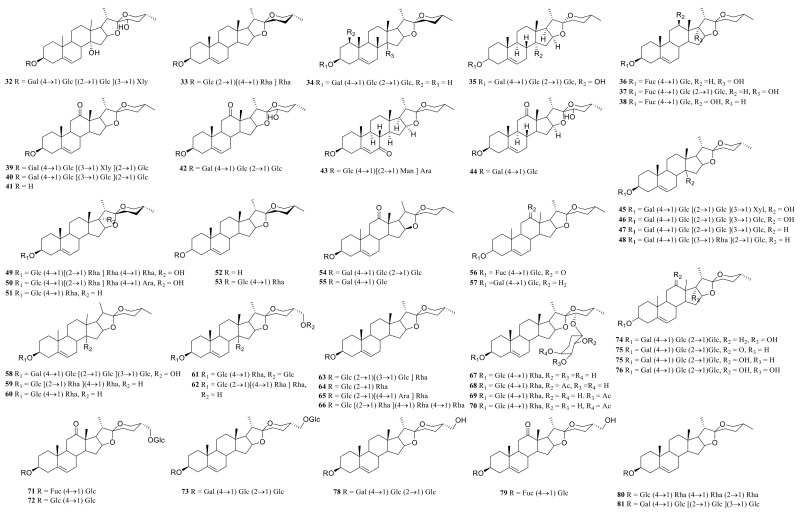
Structures of isosprirostanol from *Polygonatum* Mill.

**Figure 3 molecules-27-04821-f003:**
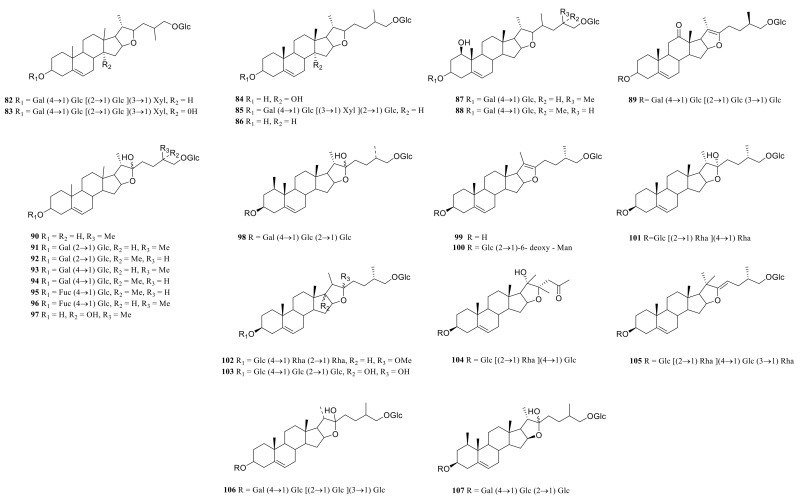
Structures of furostanol from *Polygonatum*.

**Figure 4 molecules-27-04821-f004:**
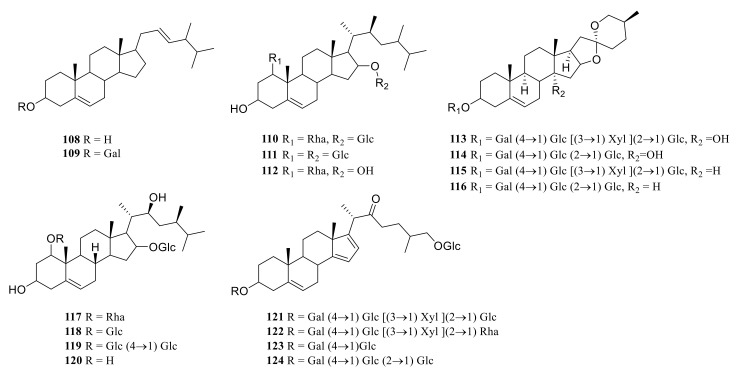
Structures of other steroidal saponin from *Polygonatum*.

**Figure 5 molecules-27-04821-f005:**
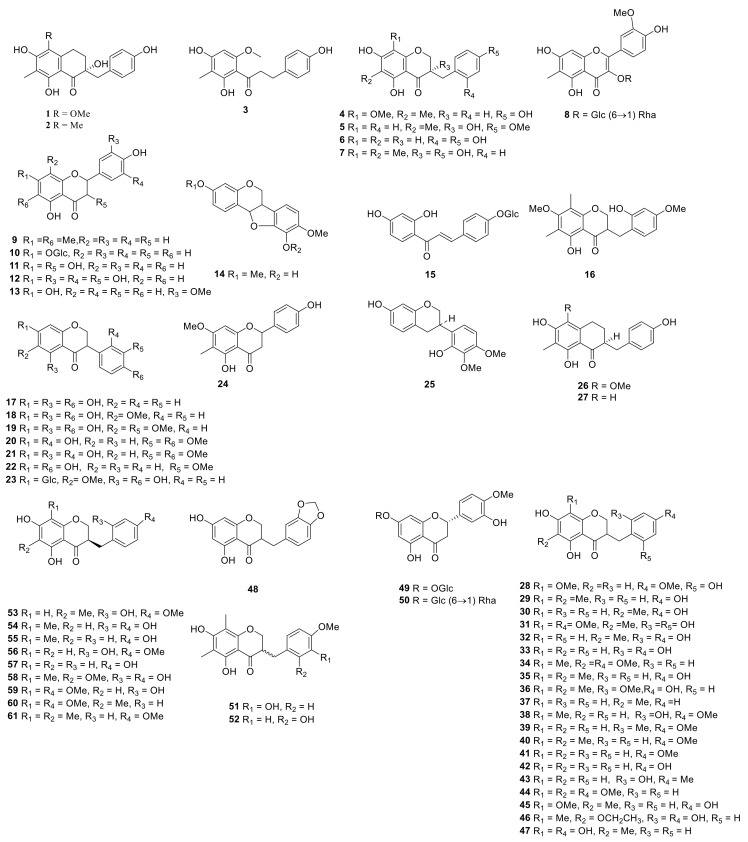
Structures of flavonoid from *Polygonatum*.

**Figure 6 molecules-27-04821-f006:**
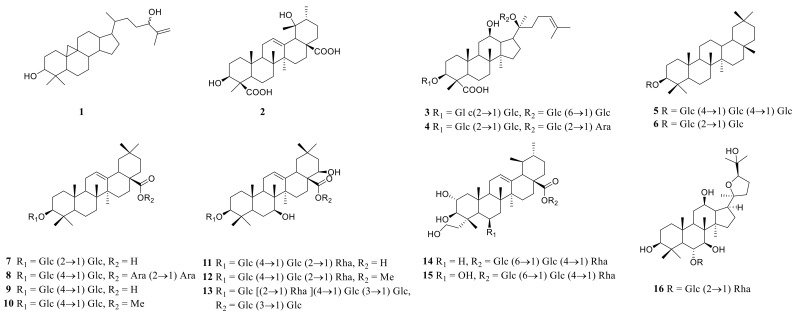
Structures of triterpenoid saponin from *Polygonatum*.

**Figure 7 molecules-27-04821-f007:**
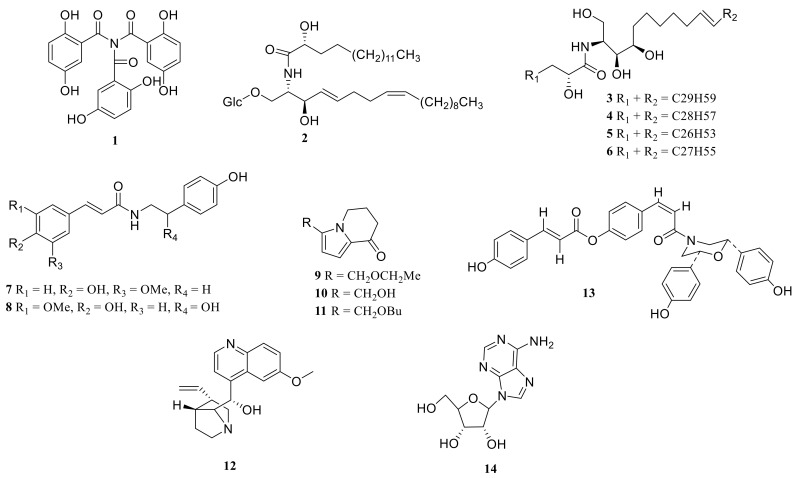
Structures of alkaloid from *Polygonatum*.

**Figure 8 molecules-27-04821-f008:**
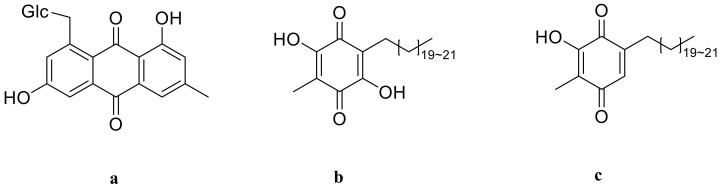
Structures of quinone from *Polygonatum*, (**a**) emodin-8-O-β-D-glucopyranoside; (**b**) polygonaquinone A; (**c**) polygonaquinone B.

**Table 1 molecules-27-04821-t001:** Species of the genus *Polygonatum*.

Number	Species	Distribution	First Recorded Time
1	*P. acuminatifolium Kom*	Russian, China	1916
2	*P. adnatum*	China	1987
3	*P. amabile*	Japan	1892
4	*P. angelicum*	Arunachal Pradesh, Tibet	2015
5	*P. arisanense*	China (Taiwan)	1920
6	*P. autumnale*	Arunachal Pradesh	2015
7	*P. annamense*	Vietnam	2015
8	*P. azegamii*	Japan	2008
9	*P. biflorum*	Canada, United Mexican States	1817
10	*P. brevistylum*	Nepal, Darjiling	1875
11	*P. buschianum*	Krym	1979
12	*P. campanulatum*	China	2015
13	*P. cathcartii*	Nepal, China	1875
14	*P. cirrhifolium*	Himalaya, China	1839
15	*P. costatum*	Thailand	2015
16	*P. cryptanthum*	Korea, Japan	1908
17	*P. curvistylum*	Nepal, China	1892
18	*P. cyrtonema*	China	1892
19	*P. daminense*	China	2020
20	*P. desoulavyi*	Korea, Japan	1931
21	*P. domonense*	Japan	1970
22	*P. falcatum*	Korea, Japan	1859
23	*P. falcatum var. hyugaense*	Japan	1957
24	*P. falcatum var. trichosanthum*	Japan	2008
25	*P. filipes*	China	1980
26	*P. franchetii*	China	1892
27	*P. geminiflorum*	Pakistan, Himalaya	1844
28	*P. glaberrimum*	Turkey, Iran	1849
29	*P. gongshanense*	China, Myanmar	2014
30	*P. govanianum*	Pakistan, Himalaya	1839
31	*P. graminifolium*	Himalaya	1851
32	*P. grandicaule*	Korea	1998
33	*P. griffithii*	Arunachal Pradesh, Tibet	1875
34	*P. hirtellum*	China	1936
35	*P. hookeri*	Himalaya, China	1875
36	*P. humile*	Kazakhstan, Japan	1859
37	*P. inflatum*	Korea, Japan	1901
38	*P. infundiflorum*	Korea	1998
39	*P. involucratum*	Russian, Korea, Japan	1883
40	*P. jinzhaiense*	China	2000
41	*P. kingianum*	China	1890
42	*P. lasianthum*	Korea, Japan	1883
43	*P. latifolium*	Europe, Turkey	1807
44	*P. leiboense*	China	1984
45	*P. longistylum*	China	1990
46	*P. luteoverrucosum*	Arunachal Pradesh, Tibet	2015
47	*P. macranthum*	Japan	1919
48	*P. macropodum*	China	1832
49	*P. megaphyllum*	China	1966
50	*P. mengtzense*	China, Vietnam	1936
51	*P. multiflorum*	Europe, Caucasus	1785
52	*P. nervulosum*	Himalaya	1875
53	*P. nodosum*	China	1892
54	*P. odoratum*	China, Europe, Japan	1906
55	*P. omeiense*	China	1992
56	*P. oppositifolium*	Nepal, Assam	1839
57	*P. orientale*	Krym, Turkey, Iran	1807
58	*P. prattii*	China	1892
59	*P. pseudopolyanthemum*	Caucasus	1928
60	*P. pubescens*	Canada, American	1813
61	*P. punctatum*	Nepal, China	1850
62	*P. qinghaiense*	China (Qinghai)	2005
63	*P. robustum*	Korea	1917
64	*P. roseum*	Asia, China (Xinjiang)	1850
65	*P. sewerzowii*	Iran, Asia	1868
66	*P. sibiricum*	Siberia, Korea, Bhutan	1811
67	*P. singalilense*	Nepal, Bhutan	1965
68	*P. sparsifolium*	China	2002
69	*P. stenophyllum*	Russian, Korea	1859
70	*P. stewartianum*	China	1912
71	*P. tessellatum*	Assam, China	1936
72	*P. tsinlingense*	China	1949
73	*P. undulatifolium*	Arunachal Pradesh, Tibet	2018
74	*P. urceolatum*	China, Vietnam	2014
75	*P. verticillatum*	Europe, China	1785
76	*P. wardii*	Assam, Tibet	1937
77	*P. yunnanense*	China	1916
78	*P. zanlanscianense*	China	1915
79	*P. zhejiangensis*	China (Zhejiang)	1994

**Table 2 molecules-27-04821-t002:** Chemical constituents of the genus *Polygonatum*.

No.	Compounds	Species	Parts	References
1	neoprazerigeninA-3-O-β-D-lycotetraosid	*P. sibiricum*	rhizome	[7]
2	(25R)-spirost-5-ene-3β,14α-diol-3-O-β-D-glucopyranosyl-(1→2)-[β-D-glucopyranosyl-(1→3)]-Β-D-glucopyranosyl-(1→4)-β-D-galactopyranoside	*P. odoratum*	rhizome	[8]
3	(25S)-spirost-5-en-3-ol-3-O-β-D-glucopyranosyl-(1→3)-[β-D-fucopyranosyl-(1→2)]- β-D-glucopyranosyl-(1→4) -β-D-galactopyranoside	*P. verticillatum*	rhizome	[9]
4	(25S)-spirost-5-ene-3β,12β-diol-3-O-{β-D-glucopyranosyl-(1→2)-[β-D-xylopyranosyl-(1→3)]-β-D-glucopyranosyl-(1→4)} -β-D-galactopyranoside	*P. cirrhifolium*	rhizome	[10]
5	(25S)-Spirosta-5,14-diene-3β-ol-3-O-{β-D-glucopyranosyl-(1→2)-[β-D-xylopyranosyl-(1→3)]-β-D-glucopyranosyl-(1→4)} -β-D-galactopyranoside	*P. odoratum* *P. cirrhifolium*	rhizome	[10]
6	(25S)-spirost-5-en-3β-ol-3-O-α-L-rhamnose (1→2)-[α-L-rhamnose (1→4)]-β-D-Glucoside	*P. cirrhifolium*	rhizome	[10]
7	(25S)-spirost-5-ene-3β,14α-diol-3-O-{β-D-glucopyranosyl-(1→2)-[β-D-xylopyranosyl-(1→3) ]-β-D-glucopyranosyl-(1→4)}-β-D-galactopyranoside	*P. odoratum* *P. cirrhifolium*	rhizome	[10,11]
8	3-O-β-D-glucopyranosyl-(1→2)-[β-D-xylopyranosyl-(1→3)]-β-D-glucopyranosyl-(1→4)-galactopyranoside-25(S)-spirost-5(6) -en-3-ol	*P. odoratum*	rhizome	[11]
9	3-O-β-D-glucopyranosyl-(1→2)-[β-D-xylopyranosyl-(1→3)]-β-D-glucopyranosyl-(1→4)-galactopyranoside-25(S)-spirost-5(6) -en-3β, 14α-diol	*P. odoratum*	rhizome	[11]
10	neosibiricoside A	*P. sibiricum*	rhizome	[12]
11	neosibiricoside B	*P. sibiricum*	rhizome	[12]
12	neosibiricoside C	*P. sibiricum*	rhizome	[12]
13	polygoside A	*P. odoratum*	rhizome	[13]
14	(3β,14α)-3-O-β-D-glucopyranosyl-(1→2)-[β-D-xylopyranosyl-(1→3)]-β-D-glucopyranosyl-(1→4)-β-D-galactopyranoside- yamogenin	*P. odoratum*	rhizome	[13]
15	(25S)-spirost-5-ene-3β, 14α-dihydroxy	*P. odoratum*	rhizome	[13]
16	(25S)-spirost-5-en-3β-ol-3-O-β-D-glucopyranosyl-(1→4)-β-D-galactopyranoside	*P. odoratum*	rhizome	[13]
17	(25S)-spirost-5-en-3β-ol-3-O-β-D-glucopyranosyl-(1→2)-β-D-glucopyranosyl-(1→4)-β-D-galactopyranoside	*P. odoratum*	rhizome	[13]
18	polygonatumoside F	*P. odoratum*	rhizome	[14]
19	polygonatumoside D	*P. odoratum*	rhizome	[15]
20	polygonatumoside E	*P. odoratum*	rhizome	[15]
21	(25S)-spirost-5-en-3-O-β-D-glucopyranosyl-(1→4)-β-D-fucopyranosyl-3β, 17α-diol	*P. sibiricum*	rhizome	[16]
22	(25S)-spirost-5-en-3β,12β-diol-3-O-β-D-glucopyranosyl-(1→4) -β-D-fucopyranosyl	*P. sibiricum*	rhizome	[16]
23	(25S)-spiroster-5-en-12-one-3-OD-glucopyranosyl-(1→2)-O-[β-D-glucopyranosyl-(1→3)]-O-β-D-glucopyranosyl-(1→4)-β-D-galactopyranoside	*P. cyrtonema*	rhizome	[17]
24	(25S)-spirost-5-en-12-one-3-O-D-glucopyranosyl-(1→2)-O-[β-D-xylopyranosyl-(1→3)]-O-β-D-glucopyranosyl-(1→4)-β-D-galactopyranoside	*P. cyrtonema*	rhizome	[18]
25	(25S) -3-β-hydroxy-spirost-5-en-12-one	*P. cyrtonema*	rhizome	[18]
26	25S-pratioside D_1_	*P. kingianum*	rhizome	[19]
27	25S-Yunnan *Polygonatum* A	*P. kingianum*	rhizome	[19]
28	(25S)-spirost-5-ene-3β,14α-diol-3-O-β-D-glucopyranosyl-(1→2)-[β-D-glucopyranosyl- (1→3)]-β-D-glucopyranosyl-(1→4) -β-D-galactopyranoside	*P. odoratum*	rhizome	[7]
29	(25S)-spirost-5-en-3β-ol-3-O-β-D-glucopyranosyl-(1→2)-[β-D-glucopyranosyl-(1→3)]-β-D-glucopyranosyl-(1→4)-β-D-galactopyranoside	*P. odoratum*	rhizome	[7]
30	(25S)-spirost-5-en-3β-ol-3-O-β-D-glucopyranosyl-(l→2) -[β-D-xylopyranosyl-(l→3) ]-β-D-glucopyranosyl-(l→4)-β-D-galactopyranoside	*P. odoratum*	Fresh rhizome	[20]
31	kingianoside H	*P. kingianum*	rhizome processed	[21]
32	sibiricoside B	*P. sibiricum*	rhizome	[7]
33	(25R)-spirost-5-en-3β-ol-3-O-α-L-rhamnose (1→2)-[α-L-rhamnose (1→4)]-β-D- Glucoside	*P. cirrhifolium*	rhizome	[10]
34	neosibiricoside D	*P. sibiricum*	rhizome	[12]
35	polygoside B	*P. odoratum*	rhizome	[13]
36	(25R)-spirost-5-en-3β,17α-diol-3-O-β-D-glucopyranosyl-(1→4)-β-D-fucopyranosyl	*P. sibiricum*	rhizome	[17]
37	(25r)-spirost-5-en-3β,17α-diol-3-O-β-D-glucopyranosyl-(1→4)-β-D-fucopyranosyl	*P. sibiricum*	rhizome	[17]
38	(25R)-spirost-5-en-3β,12β-diol-3-O-β-D-glucopyranosyl-(1→4)-β-D-fucopyranosyl	*P. sibiricum*	rhizome	[18]
39	(25R) Spiroster-5-en-12-one-3-OD-glucopyranosyl-(1→2)-O-[β-D-xylopyranosyl-(1→3)]-O-β-D-glucopyranosyl-(1→4) -β-D-galactopyranoside	*P. cyrtonema*	rhizome	[19]
40	cyrtonemoside A	*P. cyrtonema*	rhizome	[22]
41	(25r)-3-β-hydroxy-spirost-5-en-12-one	*P. cyrtonema*	rhizome	[23]
42	(25r) -kingianoside G	*P. kingianum*	rhizome	[24]
43	kingianoside K	*P. kingianum*	rhizome processed	[25]
44	kingianoside I	*P. kingianum*	rhizome processed	[25]
45	(25R)-spirost-5-ene-3β,14α-diol-3-O-β-D-glucopyranosyl-(1→2)-[β-D-xylopyranosyl-(1→3)]-β-D-glucopyranosyl-(1→4)-β-D-galactopyranoside	*P. odoratum*	rhizome	[9]
46	(25R)-spirost-5-ene-3β,14α-diol-3-O-β-D-glucopyranosyl-(1→2)-[β-D-glucopyranosyl-(1→3)]-Β-D-glucopyranosyl-(1→4) -β-D-galactopyranoside	*P. odoratum*	rhizome	[9]
47	(25R)-spirost-5-en-3β-ol-3-O-β-D-glucopyranosyl-(1→2)-[β-D-glucopyranosyl-(1→ 3)]- β-D-glucopyranosyl-(1→4)-β-D-galactopyranoside	*P. odoratum*	rhizome	[9]
48	(25R)-spirost-5-en-3β-ol-3-O-β-D-glucopyranosyl-(l→2)-[β-D-xylopyranosyl-(l→3)] -β-D-glucopyranosyl-(l→4) -β-D-galactopyranoside	*P. odoratum*	Fresh rhizome	[21]
49	saponin Tg	*P. kingianum*	rhizome processed	[23]
50	polygonatoside C_1_	*P. kingianum*	rhizome processed	[23]
51	ophiopogonin C’	*P. kingianum*	rhizome processed	[23]
52	Diosgenin	*P. cirrhifolium*	rhizome	[10]
53	(25R)-spirost-5-en-3β-ol-3-O-α-L-rhamnose (1→4)-β-D-glucoside	*P. cirrhifolium*	rhizome	[25]
54	pratioside D_1_	*P. prattii* *P. kingianum*	rhizome	[23]
55	kingianoside A	*P. kingianum*	rhizome	[24,26]
56	kingianoside B	*P. kingianum*	rhizome	[26]
57	funkioside C	*P. kingianum*	rhizome	[26]
58	(25R)-spirost-5-ene-3β,14α-diol-3-O-β-D-glucopyranosyl-(1→2)-[β-D-glucopyranosyl-(1→3)]-Β-D-glucopyranosyl-(1→4) -β-D-galactopyranoside	*P. odoratum*	rhizome	[6]
59	Dioscin	*P. kingianum* *P. punctatum* *P. cirrhifolium* *P. zanlanscianense*	rhizome, rhizome processed	[24,25,26,27,28,29]
60	Dracaenoside F	*P. cirrhifolium*	roots and rhizomes	[28,30]
61	polygonatoside D	*P. zanlanscianense*	rhizome	[29]
62	Isobalanin-3-O-α-L-rhamnopyranosyl-(1→2)- [α-L-rhamnopyranosyl-(1→4)]-β-D-pyranosyl Glucopyranoside	*P. zanlanscianense*	rhizome	[29]
63	saponin Pa	*P. kingianum*	rhizome processed	[24]
64	prosapogenin A of dioscin	*P.* *punctatum*	rhizome	[27]
65	gracillin	*P. zanlanscianense*	rhizome	[30]
66	parissaponin Pb	*P. zanlanscianense*	rhizome processed	[29]
67	polypunctoside A	*P. punctatum*	rhizome	[27]
68	polypunctoside B	*P. punctatum*	rhizome	[27]
69	polypunctoside C	*P. punctatum*	rhizome	[27]
70	polypunctoside D	*P. punctatum*	rhizome	[27]
71	polygonatoside A	*P. zanlanscianense*	rhizome	[29]
72	polygonatoside B	*P. zanlanscianense*	rhizome	[29]
73	pratioside C	*P. prattii*	root	[31]
74	pratioside A	*P. prattii*	root	[31]
75	pratioside D_1_	*P. prattii*	root	[31]
76	pratiosides E_1_	*P. prattii*	root	[31]
77	pratiosides F_1_	*P. prattii*	root	[31]
78	isonarthogenin-3-O-β-D-glucopyranosyl-(1→2)-β-D- glucopyranosyl-(1→4) -β-D-galactopyranoside	*P. zanlanscianense*	rhizome	[32]
79	polygonatoside C	*P. zanlanscianense*	rhizome	[32]
80	saponin Tb	*P. kingianum*	rhizome processed	[33]
81	odospiroside	*P. odoratum*	rhizome	[34]
82	sibiricoside A	*P. sibiricum*	rhizome	[7]
83	sibiricogenin-3-O-β-lycotetraoside	*P. sibiricum*	rhizome	[7]
84	polygonatumoside G	*P. odoratum*	rhizom	[15]
85	timosaponin H_1_	*P. odoratum*	rhizom	[15]
86	(25S) -funkioside B	*P. odoratum*	rhizom	[15]
87	25R-22 Hydroxy-curvetoxin C	*P. kingianum*	rhizom	[20]
88	22-Hydroxy-curvetoxin C	*P. kingianum*	rhizom	[20]
89	kingianoside Z	*P. sibiricum*	rhizome	[35]
90	22-Hydroxy-25(R)-furost-5-en-12-one-3β,22,26-triol-26-O-β-D-glucopyranoside	*P. odoratum*	rhizome	[36]
91	kinginaoside E	*P. kingianum*	rhizome	[20]
92	25S-kinginaoside E	*P. kingianum*	rhizome	[20]
93	25S-kinginaoside C	*P. kingianum*	rhizome	[20]
94	25S-kinginaoside D	*P. kingianum*	rhizome	[20]
95	kingianoside C	*P. kingianum*	rhizome	[20]
96	kingianoside D	*P. kingianum*	rhizome	[20]
97	saponin Pb	*P. kingianum*	rhizome processed	[25]
98	25S-kinginaoside F	*P. kingianum*	rhizome	[20]
99	3β,26-diol-25(R)-Δ5,20(22)-diene-furosta-26-O-β- D-glucopyranoside	*P. odoratum*	fresh rhizome	[22]
100	(3β,23ξ, 25R)-3-{[2-O-(6-deoxy-α-L-mannopyranosyl) -β-D-glucopyranosyl]-oxy}-22-hydroxy-furost-5-en-26-yl-β-D- glucopyranoside	*P. punctatum*	rhizome	[28]
101	protodioscin	*P. punctatum*	rhizome	[28]
102	26-β-D-glucopyranosyl-22-methoxy-(25R) -furost-5-en-3β, 26-diol-3-O-[α-L-rhamnopyranosyl-(1→2)][α-L- rhamnopyranosyl-(1→4)]-β-D-glucopyranoside	*P. zanlanscianense*	root	[22]
103	pratioside B	*P. prattii*	roots	[31]
104	polygonoide A	*P. sibiricum*	rhizome	[33]
105	polygonoide B	*P. sibiricum*	rhizome	[33]
106	22-hydroxy-25(S)-furost-5-en-12-one-3β,22,26-triol-26-O-β-D -glucopyranoside	*P. odoratum*	rhizome	[35]
107	kingianoside F	*P. kingianum*	rhizome	[36]
108	ergosta-7, 22-diene-3β, 5α, 6β-triol	*P. odoratum*	rhizome	[14]
109	(22S)-cholest-5-ene-1β,3β,16β,22-tetrol-1-O-α-L- rhamnopyranosyl-16-O-β-D-glucopyranoside	*P. odoratum*	rhizome	[14]
110	(22S)-cholest-5-ene-1β,3β,16β,22-tetrol-1,16-di-O-β-D- glucopyranoside	*P. odoratum*	rhizome	[15]
111	(25S)-3β,14α-dihydroxy-spirost-5-ene-3-O-β-D-glucopyranosyl-(1→2)-[β-D-xylopyranosyl-(1→3)]-β-D-glucopyranosyl-(1→4)-β-D-galacopyranoside	*P. odoratum*	rhizome	[15]
112	(25S)3β,14α-dihydroxy-spirost-5-ene-3-O-β-D-glucopyranosyl-(1→2)-β-D-glucopyranosyl-(1→4)-β-D-galacopyranoside	*P. odoratum*	rhizome	[15]
113	3-O-β-D-glucopyranosyl-(1→2)-[β-D-xylopyranosyl-(1→3)]-β-D-glucopyranosyl-(1→4)-β-D-galacopyranoside-yamogenin	*P. odoratum*	rhizome	[15]
114	(22S)-cholest-5-ene-1β,3β,16β,22-tetrol-1-O-α-L-rhamnopyranosyl-16-O-β-D-glucopyranoside	*P. odoratum*	rhizome	[15]
115	polygonatumoside A	*P. odoratum*	rhizome	[16]
116	polygonatumoside B	*P. odoratum*	rhizome	[16]
117	polygonatumoside C	*P. odoratum*	rhizome	[16]
118	3-O-β-D-glucopyranosyl(1→4)-β-D-fucopyranosyl-(25R)-spirost-5-en-3β,17α-diol	*P. sibiricum*	rhizome	[37]
119	3-O-β-D glucopyranosyl (1→4)-β-D-fucopyranosyl-(25S)-spirost-5-en-3β	*P. sibiricum*	rhizome	[37]
120	17α-diol (2), 3-O-β-D-glucopyranosyl(1→2)-β-D-glucopyranosyl (1→4)-β-D- fucopyranosyl-(25R)-spirost-5-en-3β,17α-diol	*P. sibiricum*	rhizome	[37]
121	3-O-β-D glucopyranosyl(1→4)-β-D-fucopyranosyl-(25R/S)-spirost-5-en- 3β,12β-diol	*P. sibiricum*	rhizome	[37]
122	(25S)-spirost-5-en-3 -ol 3-O-β-D-glucopyranosyl-(1→3)- [β-D fucopyranosyl-(1→2)]-β-D-glucopyranosyl-(1→4)-β-D-galactopyranoside	*P. verticillatum*	rhizome	[38]
123	26-O-β-D-glucopyranosyl-22ξ-hydroxy-(25R)-furost-5-en-3β, 26-diol, 3-O-β [xylopyranosyl (1→3) α-L-rhamnopyranosyl (1→2) β-D-glucopyranoside]	*P. verticillatum*	rhizome	[39]
124	3-O-β-D-xylopyranosyl (1→3) α-L-rhamnopyranosyl (1→3) β-D-glucopyranoside diosgenin	*P. verticillatum*	rhizome	[39]

**Table 3 molecules-27-04821-t003:** Flavonoids of *Polygonatuml*.

No.	Compounds	Species	Parts	References
1	polygonatone B	*P. odoratum*	rhizome	[13]
2	polygonatone C	*P. odoratum*	rhizome	[13]
3	polygonatone D	*P. odoratum*	rhizome	[13]
4	(25S)-spirost-5-ene-3β,12β-diol-3-O-{β-D-glucopyranosyl-(1→2)-[β-D-xylopyranosyl-(1→3)]-β-D-glucopyranosyl-(1→4)}-β-D-galactopyranoside	*P. odoratum*	rhizome	[13]
5	(3S)-3, 5, 7-trihydroxy-6-methyl-3-(4’-methoxybenzyl) -chroma-4-one	*P. odoratum*	rhizome	[14]
6	5, 7-dihydroxy-3-(2’, 4’-dihydroxybenzyl) -chroma-4-one	*P. odoratum*	rhizome	[14]
7	(3S)-3, 5, 7-trihydroxy-6, 8-dimethyl-3-(4’-hydroxybenzyl) -chroma-4-one	*P. odoratum*	rhizome	[14]
8	isorhamnetin-3-O-(6″-O-α-L-rhamnopyransoyl) -β-D-glucopyranoside	*P. odoratum*	rhizome	[14]
9	5,4’-Dihydroxy-7-methoxy-6-methylflavonoid	*P. odoratum*	rhizome	[13]
10	Apigenin-7-O-β-D-glucoside	*P. sibiricum*	fresh rhizome	[39]
11	kaempferol	*P. sibiricum* *P. cyrtonema*	fresh rhizome	[39]
12	myricetin	*P. sibiricum*	fresh rhizome	[39]
13	chrysoeriol	*P. odoratum*	rhizome	[40]
14	(6aR, 1laR)-10-hydroxy-3,9-dimethoxy pterostane	*P. kingianum*	rhizome	[41]
15	neoisoliquiritin	*P. kingianum*	rhizome	[41]
16	5-hydroxy-7-methoxy-6, 8-dimethyl-3-(2’-hydroxy-4’-methoxybenzyl) -chroma-4-one	*P.* *cyrtonema*	rhizome	[42]
17	5, 7, 4’-trihydroxy isoflavone	*P. odoratum*	rhizome	[14]
18	5, 7, 4’-trihydroxy-6-methoxy isoflavone	*P. odoratum*	rhizome	[14]
19	5, 7, 4’-trihydroxy-6, 3’-dimethoxy isoflavone	*P. odoratum*	rhizome	[14]
20	2’, 7-Dihydroxy-3’, 4’-Dimethoxyisoflavan	*P. kingianum*	rhizome	[41]
21	isoliquiritin	*P. kingianum* *P. alternicirrhosum*	Rhizome	[41] [43]
22	4’,7-Dihydroxy-3’-Methoxy Isoflavone	*P. kingianum*	rhizome	[43]
23	tectoridin	*P. odoratum*	root	[44]
24	liquiritigenin	*P. kingianum* *P. alte-lobatum* *P. odoratum*	rhizome	[41,43] [45]
25	isomucronulatol	*P. kingianum*	rhizome	[46]
26	(3R)-5, 7-dihydroxy-6-methyl-3-(4’-hydroxybenzyl) -chroma-4-one	*P. odoratum*	rhizome	[47]
27	(3R)-5,7-Dihydroxy-6-methyl-8-methoxy-3-(4’-hydroxybenzyl)-chroman-4-one	*P. odoratum*	rhizome	[47]
28	polygonatone A	*P. odoratum*	rhizome	[13]
29	(3R)-5,7-Dihydroxy-6,8-dimethyl-3-(4’-hydroxybenzyl)-chroman-4-one	*P. odoratum*	rhizome	[13]
30	(3R)-5,7-Dihydroxy-6-methyl-3-(4’-hydroxybenzyl)-chroman-4-one	*P. odoratum*	rhizome	[13]
31	5,7-Dihydroxy-6-methyl-8-methoxy-3-(4’-methoxybenzyl)-chroman-4-one	*P. odoratum*	root	[13]
32	5,7-Dihydroxy-6-methyl-3-(2’,4’-dihydroxybenzyl)-chroman-4-one	*P.* *cyrtonema*	rhizome	[39]
33	disporopsin	*P. odoratum*	rhizome	[40]
34	5,7-Dihydroxy-6-methoxy-8-methyl-3-(4’-methylbenzyl)-chroman-4-one	*P. odoratum*	rhizome	[40]
35	5,7-Dihydroxy-6,8-dimethyl-3-(4’-hydroxybenzyl)-chroman-4-one	*P. cyrtonema* *P. alte-lobatum* *P. odoratum*	rhizome rhizome rhizome	[42] [48] [49]
36	5, 7-dihydroxy-6, 8-dimethyl-3-(2’-methoxy-4’-hydroxybenzyl) -chroma-4-one	*P. cyrtonema*	rhizome	[42]
37	5, 7-dihydroxy-6-methyl-3-(4’-hydroxybenzyl) -chroma-4-one	*P. cyrtonema*	rhizome	[42]
38	5, 7-dihydroxy-8-methyl-3-(4’-hydroxybenzyl) -chroma-4-one	*P. cyrtonema*	rhizome	[42]
39	5, 7-dihydroxy-6-methyl-3-(4’-methoxybenzyl) -chroma-4-one	*P. cyrtonema*	rhizome	[42]
40	5, 7-dihydroxy-6, 8-dimethyl-3-(4’-methoxybenzyl) -chroma-4-one	*P. cyrtonema*	rhizome	[42]
41	5, 7-dihydroxy-3-(4’-methoxybenzyl) -chroma-4-one	*P. cyrtonema*	rhizome	[42]
42	5, 7-dihydroxy-3-(4’-hydroxybenzyl) -chroma-4-one	*P. kingianum P. cyrtonema*	rhizome	[12] [42]
43	5, 7-dihydroxy-3-(2’-hydroxy-4’-methoxybenzyl) -chroma-4-one	*P. cyrtonema*	rhizome	[42]
44	methylophiopogonanone B	*P. odoratum*	root	[44]
45	5,7-Dihydroxy-6-methyl-8-methoxy-3-(4’-hydroxybenzyl)-chroman-4-one	*P. odoratum*	root	[44]
46	ophiopogonanone E	*P. odoratum*	root	[44]
47	(3R)-5,7-dihydroxy-8-methoxy-3-(4-methoxybenzyl)-6-methylchrom-an-4-one	*P. odoratum*	rhizome	[50]
48	6-Methyl-4’,5,7-trihydroxy homoisoflavanone	*P. odoratum*	rhizome	[49]
49	5,7-Dihydroxy-6-methoxy-8-methyl-3-(2’,4’-dihydroxybenzyl)-chroman-4-one	*P. odoratum*	rhizome	[50]
50	(3R)-5,7,8-trihydroxy-3-(4-hydroxybenzyl) -6-methyl-chroma-4-one	*P. odoratum*	rhizome	[50]
51	5,7-Hydroxy-8-methoxy-3-(3’,4’-methylenedioxybenzyl)-chroman-4-one (Methyl Ophiopogon flavanone A)	*P. cyrtonema*	aboveground	[49]
52	neoliquiritin	*P. kingianum*	rhizome	[41]
53	hesperidin	*P. odoratum*	root	[44]
54	(±) 5, 7-dihydroxy-6, 8-dimethyl-3-(3’-hydroxy-4’-methoxybenzyl) -chroma-4-one	*P. odoratum*	root	[44]
55	(±) 5, 7-dihydroxy-6, 8-dimethyl-3-(2′-hydroxy-4′-methoxybenzyl) -chroma-4-one	*P. odoratum*	root	[44]
56	(3R)-5, 7-dihydroxy-6-methyl-3-(2′-hydroxy-4′-methoxybenzyl) -chroma-4-one	*P. cyrtonema*	rhizome	[42]
57	(3R)-5, 7-dihydroxy-8-methyl-3-(2′, 4′-dihydroxybenzyl) -chroma-4-one	*P. odoratum*	rhizome	[46]
58	(3R)-5, 7-dihydroxy-8-methyl-3-(4′-hydroxybenzyl) -chroma-4-one	*P. odoratum*	rhizome	[46]
59	(3R)-5, 7-dihydroxy-3-(2′-hydroxy-4′-methoxybenzyl) -chroma-4-one	*P. odoratum*	rhizome	[46]
60	(3R)-5, 7-dihydroxy-3-(4′-hydroxybenzyl) -chroma-4-one	*P. odoratum*	rhizome	[46]
61	(3R)-5, 7-dihydroxy-6-methoxy-8-methyl-3-(2′, 4′-dihydroxybenzyl) -chroma-4-one	*P. odoratum*	rhizome	[46]
62	(3R)-5, 7-dihydroxy-8-methoxy-3-(2′-hydroxy-4′-methoxybenzyl) -chroma-4-one	*P. odoratum*	rhizome	[46]
63	(3R)-5, 7-dihydroxy-6-methyl-8-methoxy-3-(4′-methoxybenzyl)-chroma-4-one	*P. odoratum*	rhizome	[46]
64	6, 8-dimethyl-5, 7-dihydroxy-3-(4′-methoxybenzyl)	*P. odoratum*	rhizome	[51]
65	5, 7-dihydroxy-3-(4′-hydroxybenzylidene) -chroma-4-one	*P. cyrtonema*	rhizome	[42]
66	€ 5, 7-dihydroxy-6, 8-dimethyl-3-(3, 4-dihydroxybenzylidene) -chroma-4-one	*P. odoratum*	rhizome	[44]
67	€ -7-O-β-D-glucopyranoside-5-hydroxy-3-(4′-hydroxybenzylidene) -chroma-4-one	*P. odoratum*	root	[44]
68	€ 5, 7-dihydroxy-8-methoxy-6-methyl-3-(3, 4-dihydroxybenzylidene) -chroma-4-one	*P. odoratum*	rhizome	[52]

**Table 4 molecules-27-04821-t004:** Triterpenoid saponins of *Polygonatum*.

No.	Compounds	Species	Parts	References
1	(24R/S)-9,19-CycloAltin-25-ene-3β,24-diol	*P. odoratum*	rhizome	[10]
2	3β, 19α-dihydroxy-12-en-24, 28-dioic acid	*P. odoratum*	rhizome	[14]
3	ginsenoside Rb1	*P. kingianum*	rhizome processed	[23]
4	ginsenoside Rc	*P. kingianum*	rhizome processed	[25]
5	β(OH)-(3→1) glucose-(4→1) glucose-(4→1) glucose-oleanane	*P. sibiricum*	rhizome	[53]
6	3β(OH)-(3→1) glucose-(2→1) glucose-oleanolic acid	*P. sibiricum*	rhizome	[53]
7	3β(OH)-(3→1) glucose-(4→1) glucose-(28→1) arabinose-(2→1) arabinose-oleanolic acid	*P. sibiricum*	rhizome	[53]
8	β, 30β(OH) 2-(3→1) glucose-(2→1) glucose-oleanane	*P. sibiricum*	rhizome	[53]
9	polygonoide C	*P. sibiricum*	rhizome	[54]
10	polygonoide D	*P. sibiricum*	rhizome	[54]
11	polygonoides C	*P. sibiricum*	rhizome	[54]
12	polygonoides D	*P. sibiricum*	rhizome	[54]
13	polygonoides E	*P. sibiricum*	rhizome	[54]
14	2β, 3β, (OH) 2-(28→1) glucose-(6→1) glucose-(4→1) rhamnose-ursic acid (asiaticoside)	*P. sibiricum*	rhizome	[53,55]
15	2β, 3β, 6β, (OH) 3-(28→1) glucose-(6→1) glucose-(4→1) rhamnose-ursic acid Oxalin)	*P. sibiricum*	rhizome	[53]
16	Pseudoginsenoside F_11_	*P. kingianum*	rhizome	[55]

**Table 5 molecules-27-04821-t005:** Alkaloids of *Polygonatum*.

No.	Compounds	Species	Parts	References
1	N, N-bis(2,5-dihydroxybenzoyl)-2,5-dihydroxybenzamide	*P. cirrhifolium*	rhizome	[10]
2	soyacerebroside II	*P. odoratum*	rhizome	[38]
3	*Polygonatum* sphingolipid A	*P. kingianum*	rhizome	[38]
4	*Polygonatum* sphingolipid B	*P. kingianum*	rhizome	[38]
5	*Polygonatum* sphingolipid C	*P. kingianum*	rhizome	[38]
6	*Polygonatum* sphingolipid D	*P. kingianum*	rhizome	[38]
7	N-trans-feruloyltyramine	*P. odoratum*	rhizome	[40]
8	N-trans-feruloyloctopamine	*P. odoratum*	rhizome	[40]
9	3-methoxyethyl-5,6,7,8-tetrahydro-8-indolinone	*P. sibiricum P. kingianum*	rhizome rhizome	[56] [46]
10	3-ethoxymethyl-5,6,7,8-tetrahydroindolizin-8-one	*P. sibiricum*	rhizome	[57]
11	kinganone	*P. kingianum*	rhizome	[46]
12	quinine	*P. verticillatum*	rhizome	[38]
13	polygonapholine	*P. alte-lobatum*	rhizome	[58]
14	adenosine	*P. sibiricum*	rhizome	[59]

**Table 6 molecules-27-04821-t006:** Processing of *Polygonatum*.

Processing Method	Auxiliary Dosage	Bibliography Source	References
If you take it alone, first use boiling water to remove the bitter juice, then steam and dry nine times.	-	Ming Dynasty “*Introduction to Medicine*”	[91]
Excellently steamed and ready to eat.	-	Qing Dynasty “*Materia Medica Justice*”	[92]
Remove impurities, wash, and remove; thoroughly moisten for 1 day, steam for 8 h, simmer for 12 h, take it out, sun until semi-dry, steam again for 8 h, simmer for 12 h until black, simmering, and oily, cut into thick slices, and dry.	-	“Guangdong Province Traditional Chinese Medicine Processing Regulations” 1984	[93]
Wash, stew thoroughly, or steam with wine, cut into thick slices, and dry.	For every 100 kg of *Polygonatum*, use 20 kg of Huangjiu	“*Chinese Pharmacopoeia*” 2020	[94]
A total of 400 g of *Polygonatum* and 2 L of black beans, cooked at the same time to remove the beans; avoid ironware.	-	Ming Dynasty “*Forbidden Prescriptions in Lu Mansion*”	[91]
*Polygonatum* Mill. is boiled until it is thin; squeeze the juice to remove the residue, and add honey.	herb: honey = 7:3/4:6	Qing Dynasty “*Huizhitang Experience Prescription*”	[95]

## Data Availability

The information that supports the findings of this study is available in this article.

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
