# Peer review of "A Review of Polygonatum Mill. Genus: Its Taxonomy, Chemical Constituents, and Pharmacological Effect Due to Processing Changes"

_molecules, 2022, doi:10.3390/molecules27154821_

Round 1

Reviewer 1 Report

 The title of the Manuscript should be corrected for the better understanding of its essence, for example 'A review of Polygonatum Mill. genus: its taxonomy classification, chemical constituents, and pharmacologigal effect due to and processing changes'.

The Abstract needs to be shortened to reflect more appropriately the main results of the research. The significant check is required for the English language and style. For instance, there are some stylistic and grammatical errors in the Abstract (lines 10-31) as well as through whole text:

pharmacological activities changes after processing.

The genus Polygonatum contains 32 species, and? 233 identified bioactive  chemical compounds,

from 1970 to 2021 2022

physiological functions  pharmacological activities (such as antioxidant activities, anti-fatigue activities, and anti-inflammatory activities).

The list of keywords should be expanded.

Introduction

I propose to add the English name of Polygonatum - 'King Solomon's-seal'. And it should be mentioned that nowadays it belongs to the Asparagaceae family (not Liliaceae) https://en.wikipedia.org/wiki/Polygonatum

It would be excellent to provide data on the presence of these species in the other world Pharmacopoeias - European, American, etc.

The purpose of the article is in the abstract, but not described in the introduction section.

 The data in Table 2 would be desirable to present in a more concise form.

The name of section 5 'Discuss' is not correct. It should be Discussion. This section is too short and does not contain literary references, so it should be substantially revised.

The Conclusions need to be rewritten to reflect more succinctly the aim and main results of the research.

The italic type should be used everywhere for writing Latin names of genus and species (lines 10, 15, 16, etc).

Finally, 40 results were found by me after searching words 'Polygonatum 2022' in the scientific Database PubMed 24 June 2022. So, authors should add at minimum several of them to update the manuscript (only 6 literary sources published in 2022 were used by them).

Author Response

Point-by-point responses to the reviewers’ comments

Reviewer: 1

Minor points (partly indicated, for not all)

1.The title of the Manuscript.

Answer: The title has been changed to "A review of the genus Polygonatum: its taxonomy classification, chemical constituents, and pharmacological effects due to processing changes" as you suggested.

  1. The Abstract and some stylistic, grammatical errors.

Answer: The abstract has been shortened, and some stylistic and grammatical errors have been corrected. The “2021” was corrected as “2022”. The “changes” was corrected as “change”. The “233 chemical compounds” was corrected as “233 identified bioactive chemical compounds”, and “physiological functions” was corrected as “pharmacological activities”. Please kindly see lines 31-33 in the revised manuscript.

  1. The list of keywords.

Answer: The list of keywords was expanded.

  1. English name of Polygonatum.

Answer: The English name of Polygonatum - 'King Solomon's-seal' was added as you suggested. Please kindly see line 52 in the revised review.

  1. The presence of these species in the other world Pharmacopoeias.

Answer: Neither the “US Pharmacopoeia” nor the “European Pharmacopoeia” lists Polygonatum; only the “Chinese Pharmacopoeia”, the “Japanese Pharmacopoeia”, and the contents of Polygonatum in the “Japanese Pharmacopoeia” are the same as those in the “Chinese Pharmacopoeia”.

  1. Question about family of Polygonatum.

Answer: Since the “Chinese Pharmacopoeia” is updated every five years, I had written Liliaceae by mistake. It was corrected as the Asparagaceae family. Please kindly see line 52 in the revised manuscript.

  1. Question about Introduction.

Answer: The purpose of the article was described in the introduction. Please kindly see lines 82-87 in the revised manuscript.

  1. Section 5.

Answer: “Discuss” was corrected as “Conclusions”, and the content of conclusions had been rewritten.

  1. Question about italic type.

Answer: I am sorry for my previous negligence. All Latin names were italicized.

10.Update on the manuscript

Answer: The species of Polygonatum Mill. were updated from 73 to 79, and the manuscript was added some references published in 2022. Please kindly see the Table 2 and references in the revised manuscript.

Reviewer 2 Report

This manuscript deals with a review about chemical constituents of species belonging to genus Polygonatum, it also shows the pharmacological activities of the compounds and the changes in the chemical composition that produce the different isolation processes. The topic addressed in this manuscript is within the topics covered by the journal Molecules, and in the scope of the Special Issue: Biological Activities of Natural Products III. However, before being published there are some errors that must be corrected, some points that must be clarified and some suggestions for improvement that must be addressed.

Title

-The title is very ambiguous, considering that "chemical constituents" refers to the compounds that have been isolated from the species of this genus, "classification" could be refers to the species or the compounds, and "Processing changes" is also not clear what it refers to. A main section in this manuscript is called "3. Pharmacological activities" which is not referred to in the title. I suggest rewriting the title considering all these aspects.

Abstract

-Line 11-12: … such as……. such as…., correct.

-The "Results" and “Conclusions” sections are not shown in the body of the document.

Introduction

-Line 35: "....Asparagales Link family of the Liliaceae family.", correct. A family cannot belong to another family

-Line 20: “(2020edition), correct.

-Line 40-41. “Polygonatum kingianum Coll. et Hemsl, Polygonatum sibiricum red, and Polygonatum cyrtonema Hua are species of Polygonatum rhizome”, correct. A species cannot belong to another species.

-Line 55: "in order to reduce toxic components" how is this possible, by evaporation, by degradation? explain.

-Line 58, page 2: "changes after processing of medicinal plants" change the plants? rewrite.

-Cite Figure 1 in the text

-Line 64: There are some concordance errors throughout the manuscript, it is suggested to review and correct, for example "The genus Polygonatum is a big family with many species" a genus cannot be a family

Line 64: Consider that genus Polygonatum it is a medium genus in number of species

Chemical constituents of Polygonatum Mill.

-Line 71: "As the introduction says" I consider that it is not an adequate sentence.

-The information on lines 71-72 is repeated with that on lines 50-51.

Processing of Polygonatum Mill.

- Relate the methods: steaming, wine steaming, wine stewing, with those described in Table 6.

Line 179: P. odoratum druce should be  Polygonatum odoratum (Mill.) Druce or simply P. odoratum.

Discuss

-This section must be completely rewritten, correcting sentences such as "unification of medicinal material blending and the standardization of the medicinal material market" or "The final objective should be Polygonatum Mill. medicinal plants”. In addition, the reference to "quality" should be removed since this is not mentioned in the body of the manuscript.

-Line: 243 "the author believes" should be plural

General remarks

-Many parts of the manuscript need to be revised and rewritten, especially the introduction. As an example, the errors of only one paragraph will be indicated, in which there are errors such as:

The use of terms that do not correspond, are not defined or have no bibliographical reference, for example: line 42-44, "transitional morphology and the overlap in geographical distribution"; Line 46 "chaotic varieties of the base source"

Sentences without continuity or without concordance: line 45-46 "chaotic varieties of the base source have led to the restriction of therapeutic compounds from the genus Polygonatum"; line 47-48 "there is a mixture between the original plant cultivation and medicinal use"

-Improve resolution of figures, Figure 2 is unreadable

- Figure, Fig. or Fig, homogenize

- Correct the concordance of tenses, for example in the sentence "There are many methods of processing the genus Polygonatum in the past" “there are” is present.

-There are many initials without definition, for example "HPLC-MS/MS"

-I suggest changing the 1.1 theme. "Classification of Polygonatum Mill." to section 2 and renumber.

-The name of section 5. Discuss should be changed to Conclusions

Author Response

Reviewer 2:

  1. Question about the title of this manuscript.

Answer: Thanks for your kind comment. The title was corrected as "A review of the genus Polygonatum: its taxonomy classification, chemical constituents, and pharmacological effects due to and processing changes. "

  1. Line 11-12: … such as……. such as…., correct.

Answer: The "physiological functions (such as anti-oxidation activities) " was corrected as "pharmacological activities (such as anti-oxidant activities) ". Please kindly see lines 31-33 in the revised manuscript.

  1. The "Results" and "Conclusions" sections are not shown in the body of the document.

Answer: The "Results" and "Conclusions" sections of abstract had been rewritten. Please kindly see lines 31-40 in the revised manuscript.

  1. ".... Asparagales Link family of the Liliaceae family.", correct.

Answer: This mistake was corrected. The "… of the Liliaceace family" was deleted. Please kindly see line 52 in the revised manuscript.

  1. Line 20: (2020 edition), correct.

Answer: The (2020 edition) was corrected as (2020 edition). Please kindly see line 59 in the revised manuscript.

  1. 6. Line 40-41. “Polygonatum kingianum Coll. et Hemsl, Polygonatum sibiricum red, andPolygonatum cyrtonema Huaare species of Polygonatum rhizome”, correct.

Answer: This sentence was deleted.

  1. 7. "in order to reduce toxic components", explain.

Answer: Calcium oxalate monohydrate raphides and volatile components might be the irritating components of the genus Polygonatum. By repeated steaming or stewing, there were far fewer calcium oxalate monohydrate raphide. The raphide bundles that remained were adhered together and difficult to separate, and the majority of single raphide were disintegrated, particularly at their tips. The volatile components (such as n-hexanal and camphene) was volatilized. This explanation and references were added in the manuscript. Please kindly see the section 5.1 on pages 44-45.

  1. 8. page 2: "changes after processing of medicinal plants".

Answer: I am sorry I didn't express well. This sentence was deleted.

  1. 9. Cite Figure 1 in the text

Answer: The previous Figure 1 does not express the highlights of the entire article, so I repainted it and cited it. Please kindly see lines 322-328, page 4.

  1. 1 Line 64

Answer: "The genus Polygonatum is a big family with many species" was corrected as "The genus Polygonatum comprises 79 species". Please kindly see line 92 in the revised manuscript.

  1. 1 "As the introduction says", corrected.

Answer: "As the introduction says" was corrected as "As the introduction mentioned". Please kindly see line 104 in the revised manuscript.

  1. 1 The information on lines 71-72 is repeated with that on lines 50-51.

Answer: Sentences was rewritten. Please kindly see lines 70-71 and 104-105 in the revised manuscript.

  1. 1 Introduction, rewrite.

Answer: I'm sorry for my mistakes. The Introduction was rewritten. Please kindly see the Introduction in the revised manuscript.

  1. 1 Correct the concordance of tenses

Answer: “there are” was corrected as “there were”, Please kindly see line 321 in the revised manuscript, and I have checked elsewhere for tenses errors.

  1. 1 The name of section 5.

Answer: The name of section 5 was corrected as Conclusions. Thanks for your kind suggestion.

Reviewer 3 Report

The effort made by the authors is very valuable, as nowadays it is impossible to follow most of the current literature on a topic of interest. This paper is a comprehensive review focused on the classification, chemical composition, and changes after processing of Polygonatum species. The manuscript fits within the scope of the journal. The title is clear and it is adequate to the content of the article. The author’s work on discussing achieved results is appreciated.

I have some recommendations for authors:

What is the novelty and originality of the review, considering that recent articles on this topic are published? There are review articles, on the same topic, published in recent years.

Please check the correctness of the written information: There are over 70 species in the genus Polygonatum, not 61 as mentioned; the family is Asparagaceae (according to WorldFlora) not Liliaceae, as you mentioned...

Please include citations in the text. There are many paragraphs that are not supported by the bibliography.

Use italics for genus and species names!

Figure 1 is original? Please include this information in the figure title.

L64: ” The genus Polygonatum is a big family with many species” The phrase is confusing: The genus of plants is not a family. Maybe it would be better: The genus Polygonatum includes many species.

Check the bibliography to be written according to the requirements of the journal.

Change the Discuss chapter to Conclusions.

Author Response

Reviewer 3:

What is the novelty and originality of the review, considering that recent articles on this topic are published? There are review articles, on the same topic, published in recent years.

Answer: Thanks for your question. It is true that some articles on the same topic have been published in recent years, but they all focus on certain species (P. odoratum, P. cyrtonema, P. kingianum, and P. sibiricum). Other species like P. verticillatum, P.cirrhifolium, P.zanlanscianense also has bioactivity compounds, but little attention.

I wish this review to critically evaluate available research reports on the genus, and systematically organize and present the findings.

1.Please check the correctness of the written information: There are over 70 species in the genus Polygonatum, not 61 as mentioned; the family is Asparagaceae.

Answer: Since the “Chinese Pharmacopoeia” is updated every five years, I had written Liliaceae by mistake. It was corrected as the Asparagaceae family. Please kindly see line 52 in the revised manuscript.

  1. 2. There are over 70 species in the genus Polygonatum, not 61 as mentioned

Answer: I have updated the species information of the genus Polygonatum. The genus Polygonatum comprises many 79 species. Among them, 39 species distributed in China were recorded in the Chinese monograph “Flora of China”, and other 40 species were included in the World Checklist of Selected Plant Families (WCSPF, World Checklist of Selected Plant Families: Royal Botanic Gardens, Kew). Please kindly see Table 1 in the revised manuscript.

  1. Please include citations in the text.

Answer: I have rewritten some of the content; others have cited literature.

  1. Use italics for genus and species names.

Answer: Please forgive my negligence, All Latin names were italicized.

  1. Figure 1 is original? Please include this information in the figure title.

Answer: I redrew the Figure 1, and cited. Please kindly see line 87 in the revised manuscript.

  1. 6.“The genus Polygonatum is a big family with many species”, correct.

Answer: “The genus Polygonatum is a big family with many species” was corrected as “The genus Polygonatum comprises 79 species”, because I have already written “The genus Polygonatum includes many species.” Thanks for your suggestion.

  1. 7. Check the bibliography to be written according to the requirements of the journal.

Answer: I have checked and corrected the bibliography to be written according to the requirements of the journal. Please kindly see the references.

  1. 8.Change the Discuss chapter to Conclusions.

Answer: I changed the Discuss chapter to Conclusions as you suggested.

Round 2

Reviewer 1 Report

The title of the Manuscript should be slightly corrected, for example 'A review of Polygonatum Mill. genus: its taxonomy, chemical constituents, and pharmacological effect due to processing changes'.

The moderate check is required for the English language and style.

The italic type should be used everywhere for writing Latin names of genus and species (lines 711, 713 etc.).

The Abstract needs to be moderately corrected:

The title of the Manuscript should be slightly corrected, for example 'A review of Polygonatum Mill. genus: its taxonomy, chemical constituents, and pharmacological effect due to processing changes'.

The moderate check is required for the English language and style.

The italic type should be used everywhere for writing Latin names of genus and species (lines 711, 713 etc.).

The Abstract needs to be slightly corrected, for instance -

Ethnopharmacological relevance: The genus Polygonatum Polygonatum Tourn. ex Mill. contains numerous chemical components such as steroidal saponins, polysaccharides, flavonoids, alkaloids, and others, it possesses diverse pharmacological activities, such as anti-aging, anti-tumor, immunological regulation, as well as blood glucose management and fat reducing properties.

Aim of the review: This study reviews the current state of research on the systematic categorization, chemical composition, pharmacological effects, and processing changes of the plants belonging genus Polygonatum, with the goal of providing a theoretical foundation for their scientific development and rational application.

Materials and methods: The information was obtained by searching the scientific literature published between 1977 and 2022 on online databases (including PubMed, CNKI, SciFinder, and Web of Science) and other sources (such as the Chinese Pharmacopoeia 2020 edition, and Chinese herbal books).

Results: The genus Polygonatum contains 79 species, and 233 bioactive chemical compounds were identified in them. The abundance of pharmacological activities such as antioxidan, anti-fatigue,, anti-inflammatory, etc. were revealed for the representatives of this genus. In addition, there are numerous processing methods, and many chemical constituents and pharmacological activities change after the unappropriated processing.

Conclusions: This review summarized the taxonomy, chemical composition, pharmacological effects, and processing of the plants belonging to the genus Polygonatum, providing references and research tendencies for plant-based drug development and further clinical applications.

Author Response

Reviewer 1

Thanks for your kind comments. We apologize for the grammatical problems and have corrected them based on your suggestions.

1 Title.

Answer: The title of the Manuscript was corrected as “A review of Polygonatum Mill. genus: its taxonomy, chemical constituents, and pharmacological effect due to processing changes”, as you suggested.

2 The italic type.

Answer: All Latin names were italicized. Please kindly see lines 344 and 345.

3 The Abstract needs to be slightly corrected.

Answer: Ethnopharmacological relevance: “Polygonatum Mill. was corrected as “Polygonatum Tourn. ex Mill.” (Line 11). Aim of the review: “the genus Polygonatum” was corrected as “the plants belonging genus Polygonatum” (Line 17), and “the genus Polygonatum’s” was corrected as “their” (Line 18). Results: “233 bioactive chemical compounds in them” was corrected as “..., and 233 bioactive chemical compounds were identified in them” (Lines 23-24). “…and the chemical constituents and pharmacological activities change after processing” was corrected as “…, and many chemical constituents and pharmacological activities change after the unappropriated processing” (Lines 27-28). Conclusions: “pharmacological effect, … the plants Polygonatum” was corrected as “pharmacological effects, … the plants belonging to the genus Polygonatum” (Lines 29-30).

Thank you again for your patience in revising this manuscript.

Reviewer 2 Report

The manuscript has been improved, errors have been corrected, and suggestions and comments have been taken into account, so I believe that it can be published in present form.

Author Response

Dear reviewer,

Thank you for your professional suggestions and patience in revising this manuscript.

Yours sincerely,

Lu Luo

Reviewer 3 Report

The manuscript has improved in its second revision. I’m still not convinced about the novelty of the work, this must be evaluated by the editor.

 Specific comments: Figure 1 does not bring new information to the manuscript.

I propose the deletion of Table 1. It has no value for the topic of the manuscript.

Author Response

Thanks for your approval for the improvement of our second manuscript.

1 About the novelty of the work.

Answer: I would like to say a few more words.

(1) In PubMed, there have been 12 reviews about the Polygonatum in recent five years, but two on the whole genus.

(2) The previous reviews were not comprehensive. One only reviewed 60 species and wrote about ethnopharmacology, phytochemistry, and pharmacology, which were published in 2018 [1]. One is research about progress methods, constituents, and pharmacological effects, which was published in a Chinese journal in 2019 [2].

(3) Plant research in this genus has developed rapidly in recent years, and the amount of literature has increased rapidly (A search of Polygonatum on PubMed yields 40 papers published in 2022.). It is necessary to make a timely summary.

2 Figure 1 does not bring new information to the manuscript.

Answer: It was deleted.

3 Table 1 has no value for the topic of the manuscript.

Answer: Thanks for your kind comments. This has been requested by other reviewers. I hope that Table 1 provides information (like number of species, distribution) on species of the genus Polygonatum, and summarizes all Polygonatum species to date.

References:

[1] Zhao P, Zhao C, Li X, Gao Q, Huang L, Xiao P, Gao W. The genus Polygonatum: A review of ethnopharmacology, phytochemistry and pharmacology. J Ethnopharmacol. 2018 Mar 25;214:274-291. doi: 10.1016/j.jep.2017.12.006. Epub 2017 Dec 12. PMID: 29246502.

[2] Zhang J, Wang YZ, Yang WZ, Yang MQ, Zhang JY. [Research progress in chemical constituents in plants of Polygonatum and their pharmacological effects]. Zhongguo Zhong Yao Za Zhi. 2019 May;44(10):1989-2008. Chinese. doi: 10.19540/j.cnki.cjcmm.20190222.006. PMID: 31355552.